# EventDTW: An Improved Dynamic Time Warping Algorithm for Aligning Biomedical Signals of Nonuniform Sampling Frequencies

**DOI:** 10.3390/s20092700

**Published:** 2020-05-09

**Authors:** Yihang Jiang, Yuankai Qi, Will Ke Wang, Brinnae Bent, Robert Avram, Jeffrey Olgin, Jessilyn Dunn

**Affiliations:** 1The Departments of Biomedical Engineering and Biostatistics & Bioinformatics, Duke University, Durham, NC 27708, USA; yh_jiang@hust.edu.cn (Y.J.); yuankai.qi@duke.edu (Y.Q.); ke.wang064@duke.edu (W.K.W.); brinnae.bent@duke.edu (B.B.); 2The Division of Cardiology and the Cardiovascular Research Institute, University of California San Francisco, San Francisco, CA 94143, USA; Robert.avarm@ucsf.edu (R.A.); jeffrey.olgin@ucsf.edu (J.O.)

**Keywords:** dynamic time warping, signal alignment, nonuniform sampling

## Abstract

The dynamic time warping (DTW) algorithm is widely used in pattern matching and sequence alignment tasks, including speech recognition and time series clustering. However, DTW algorithms perform poorly when aligning sequences of uneven sampling frequencies. This makes it difficult to apply DTW to practical problems, such as aligning signals that are recorded simultaneously by sensors with different, uneven, and dynamic sampling frequencies. As multi-modal sensing technologies become increasingly popular, it is necessary to develop methods for high quality alignment of such signals. Here we propose a DTW algorithm called EventDTW which uses information propagated from defined events as basis for path matching and hence sequence alignment. We have developed two metrics, the error rate (ER) and the singularity score (SS), to define and evaluate alignment quality and to enable comparison of performance across DTW algorithms. We demonstrate the utility of these metrics on 84 publicly-available signals in addition to our own multi-modal biomedical signals. EventDTW outperformed existing DTW algorithms for optimal alignment of signals with different sampling frequencies in 37% of artificial signal alignment tasks and 76% of real-world signal alignment tasks.

## 1. Introduction

Dynamic time warping (DTW) is a class of pattern matching algorithms for aligning temporal sequences. DTW aims to provide an optimal similarity-based match between two sequences regardless of dynamic spatiotemporal differences, searching for the optimal path by minimizing cumulative point-to-point distances. DTW allows for a flexible definition of cumulative point-to-point distances to optimize alignment performance for different datasets. Because of this flexibility, DTW is widely used in speech recognition under varying speaking speeds [1], gesture recognition [2,3], and time series clustering [4].

### 1.1. DTW for Different Sampling Frequencies

Measurement modalities utilizing multiple simultaneous sensors with different sampling frequencies and/or uneven sampling rates are common in many applications, particularly in biomedicine [5,6]. In practice, we have observed that DTW performs poorly when applied to signals that were simultaneously measured at different frequencies and/or when each individual signal has a dynamic, uneven sampling frequency. This is due to the singularity problem [7,8], which results from inaccurate matching of a single observation in one time sequence with a large number of consecutive observations in another, resulting in an uneven distribution of point-to-point alignments. The singularity problem is undesirable but common in the alignment of biomedical signals of different and dynamic sampling frequencies.

A real-world example of a singularity problem using two biomedical devices to measure the same biological phenomenon (heart rate) with two different modalities with different sampling frequencies (ECG, black (~1.26) and photoplethysmogram (PPG), blue (~0.2 Hz)) is shown in Figure 1a. Here, we explored whether accounting for local structural information when performing DTW-based signal alignment could both reduce singularity occurrences and improve alignment accuracy [8].

### 1.2. Existing DTW Improvement Methods

Various improved DTW algorithms have been developed and applied to different non-temporal datasets [9,10]. Keogh et al. developed derivative DTW (dDTW), which produces intuitively correct “feature-to-feature” alignment between two sequences by using the first derivative of time series sequences as the basis for DTW alignment. Zhao et al. developed shape DTW (sDTW), which computes similarities between observations based on their local neighborhoods instead of based only on their single-point amplitudes [11]. However, we have observed that widely used or current state-of-the-art DTW algorithms, including dDTW and sDTW, do not produce optimal alignment results with respect to the singularity problem when the signal frequencies differ. When the frequency of the companion sequence is different than that of the reference sequence, dDTW and sDTW can result in suboptimal alignments because they use information from proximal observations within the companion sequence which may not have corresponding observations in the reference signal. In this paper, we are aiming to improve the performance of the DTW algorithm in aligning unevenly down-sampled signals.

A common method employed to reduce instances of singularities involves adding penalties to the cumulative distance minimization function in the DTW algorithm [8]. Early approaches that constrain the warping path can be categorized into three types: Windowing [12], slope weighting [13], and step patterning [14]. However, these common methods are not robust because they apply a hard-coded limit to the warping path which can result in improper alignment. The parameters of these algorithms also have to be tuned empirically for each application through optimization experiments, making these methods less generalizable to new applications and different types of signals.

### 1.3. Existing DTW Evaluation Metrics

Despite extensive research on DTW algorithm development to solve the singularity problem, methods to evaluate signal alignment are currently limited. Since DTW is a matching result calculated from distance measurements but not a metric [15], we need a new metric to evaluate signal alignment quality. Often, signal alignment evaluation is performed qualitatively by visually examining spatiotemporal feature-to-feature matching. A common quantitative evaluation method is the mean deviation metric [16], but this requires evenly sampled signals with same sampling rate, which is uncommon when recording real-world biomedical signals using different types of sensors. Other quantitative metrics that have been used to evaluate signal alignment include the length of the warping path [8] and the dataset classification accuracy [17]. All previous evaluation metrics describe the quality of alignment by calculating and visualizing the warping path, without quantitatively describing the severity of the singularity problem. 

Further, the singularity problem in DTW continues to be a challenge when aligning multi-modal signals. Here we propose an improved DTW algorithm, EventDTW (eDTW). eDTW takes local structural information into account when aligning signals. Although there have been different attempts to align signals with dynamic time warping [18,19], they do not propose a metric to evaluate the alignment quality. Here, we develop two criteria to evaluate DTW algorithm performance: The error rate (ER) and the singularity score (SS). We describe here our methods and results of applying eDTW to 84 publicly available time series datasets and 53 multi-modal biomedical datasets collected by our team.

## 2. Materials and Methods

In this paper, we propose an improved DTW algorithm and evaluate its performance against existing DTW algorithms in terms of the severity of the singularity problem and the accuracy of the resulting alignment between signals. (The code is available at https://github.com/Big-Ideas-Lab/DBDP/tree/master/DigitalBiomarkers-Preprocessing.) To enable comparison between algorithms, we designed two metrics, the singularity score and the error rate, to evaluate these two aspects, respectively. Using a gold-standard open-access public dataset [20] that contains 84 different signals collected in various scenarios, we designed a warped companion in silico signal for each of the signals to simulate simultaneous measurements of the same phenomenon using different sensors. To generate these companion signals, we synthetically warped and then down sampled the reference signal R which generated the warped companion signal Q. Because R were one-dimensional signals containing only measurements of amplitude, we assumed a constant sampling frequency. However, for future applications of this work, signals with non-constant sampling frequencies may be used for wider applicability because randomness in timestamps can be easily incorporated into the eDTW formulation. We applied various DTW algorithms to align R and Q and evaluated their performance based on the error rate and singularity score. We demonstrated an improved performance of eDTW in terms of accuracy of alignment and reducing the severity and occurrences of singularities. The remainder of the methods section is explained in the following order: Companion signal generation, calculation of the optimal alignment, design of the error rate metric, design of the singularity score metric, design of the event DTW algorithm, and experiments to compare the performance of different DTW algorithms.

### 2.1. Companion Signal Generation

In order to evaluate and compare DTW algorithm performance, we required two signals with a single optimal alignment against which newly calculated alignments could be compared. To achieve this, we generated a companion signal Q based on a reference signal R, where R contained L evenly spaced observations. This signal pair generation mimicked the warping and distortion often observed between real signals from multiple sensors measuring the same phenomenon at different sampling frequencies, as shown in Figure 1b. The higher-frequency ECG (black; ~1.22 Hz) and the lower-frequency PPG (blue, 0.5 Hz) were measured simultaneously, and warping can be seen in a portion of the PPG signal that is highlighted by the red rectangle. We denoted the reference signal sequence as R, the warped signal as $R^{\prime }$, and the warped and down-sampled companion signal as Q.

We described a single observation in a signal in terms of the temporal variable t (which is typically the x-axis timestamp) and the amplitude v (which is typically the value of the measurement). For example, the $i^{th}$ observation in R is given as $R_{i}\left(t_{R_{i}},\;v_{R_{i}}\right)$. Given the constant sampling frequency of R, the time interval between each adjacent observation in R is one time unit. Therefore, the timestamp of R is given by:(1)$$t_{R_{i}}=i,i\in \left\{0,1,\ldots ,L-1\right\}$$

In order to generate the warped and down-sampled companion signal Q, we first generated an intermediate signal $R^{\prime }$ and then obtained Q following the two steps.

#### 2.1.1. Generating $R^{\prime }$

We chose an anchor point at the center of R to act as the center of the synthetic warping process. This helped to prevent the warping from occurring outside of the signal bounds. The anchor point was denoted by $R_{A}\left(t_{R_{A}},\;v_{R_{A}}\right)$ where $t_{R_{A}}=L/2$.

To generate the temporal variable $t_{R_{i}^{\prime }}$ for $R^{\prime }$, we introduced a time distortion to R by warping the temporal variable t. We denoted the magnitude of the horizontal shift for the $i^{th}$ observation’s timestamp as $s_{i}.$ Therefore, the timestamp of each observation in $R^{\prime }$ is given by: (2)$$t_{R_{i}^{\prime }}=i-s_{i},i\in \left\{0,1,\ldots ,L-1\right\}$$

In order to fully contain the entire warping process within the signal bounds, we designed the value for the shift $s_{i}$ to be dynamic over the length of the signal, with the strongest shift closest to the anchor point, $R_{A}$, and nearly zero at the ends of the signal, as shown in Figure 2a, top. To realize this feature, we defined $s_{i}\;$ to be inversely proportional to the distance between $R_{i}$ and $R_{A}\;$ using the derivative of the sigmoid function:(3)$$s_{i}=\frac{e^{-\left|t_{R_{i}}-t_{R_{A}}\right|/w}}{{\left(1+e^{-\left|t_{R_{i}}-t_{R_{A}}\right|/w}\right)}^{2}}$$

In comparison with the inverse squared distance $1/{\left(t_{Ri}-t_{R_{A}}\right)}^{2}$ employed by (Keogh 2001) for the same purpose, $s_{i}$ descended from the anchor point in smoother flanks (Appendix A). The parameter w was a constant value proportional to $L_{R}$ and it defined the warping window width. For w < 0.5, the warping process was contained within the signal bounds. The smaller the value of w, the narrower the warping process. By defining w as 10% of the length of R, we localized the extreme warping around the anchor point $R_{A}$ while keeping other parts of the signal relatively unchanged (Figure 2a, middle).

To generate the amplitude variable $v_{R_{i}^{\prime }}$ for $R^{\prime }$, we introduced an amplitude distortion to R by warping the amplitude variable v. We used a Gaussian bump (Figure 2a, bottom) as the magnitude of the vertical shift for the $i^{th}$ observation’s timestamp as below:(4)$$v_{R\prime_{i}}=\{\begin{array}{cc} v_{R_{i}}+1/e^{{\left[\left(t_{Q_{i}}-t_{R_{A}}\right)/w\right]}^{2}-1}, & \left|t_{Q_{i}}-t_{R_{A}}\right|<w\\ v_{R_{i}}, & otherwise \end{array}$$

#### 2.1.2. Generating Q

To generate the down-sampled companion signal Q from the warped signal $R^{\prime }$, we calculated the timestamps and amplitudes and defined each observation in Q as $Q_{j}$($t_{Q_{j}}$, $v_{Q_{ij}}$).

By our definition, the timestamp of Q was down-sampled from that of R such that their timestamps, $t_{Q_{j}}$ and $t_{Ri},$ were stably related through the down-sampling parameter, d:(5)$$t_{Q_{j}}=t_{R_{\left[i^{*}d\right]}},\;i\in \left\{1,\ldots ,\frac{L}{d}\right\}$$

The result was that the timestamp of the $j^{th}$ observation in Q was to equal the ${\left(i^{*}d\right)}^{th}$ observation in R. Using this relationship between the timestamp of Q and the timestamp of R, we linearly interpolated the missing amplitude values of the warped signal $R^{\prime }$ that would have occurred at timestamp $t_{R_{i}^{\prime }}=t_{Q_{j}}$. These interpolated amplitude values were used to define $v_{Q_{j}}$. As seen in Figure 2b, the orange dotted line shows the time shift between $R_{i}$ and ${R^{\prime }}_{i}$, and the black dotted line shows that the timestamps of $Q_{j}$ are the same as that of $R_{\left[i*d\right]}$. An example of the original signal R, the intermediate signal $R^{\prime }$, and the warped and down-sampled companion signal Q is shown in Figure 2c.

### 2.2. Calculation of the Optimal Alignment

In DTW, point-to-point links were made between every observation in Q and at least one observation in R. Given that observations in R and Q were evenly distributed and that the down-sampling ratio was d, each observation in Q was optimally aligned with d observations in R. We first calculated an intermediate alignment between a single observation in Q and  d observations in $R^{\prime }$ by calculating the minimum Euclidean distance. Since each observation in $R^{\prime }$ has a corresponding observation with the same index in R, we could subsequently find the optimal alignment and we denoted this as: $Q_{j}\overset{opt}{↔}G_{j},$ where $G_{j}=\left\{Q_{j},\;R_{i},\;R_{i+1},\ldots ,\;R_{i+d-1}\right\}$, meaning that $G_{j}$ contains $Q_{j}$ and a subgroup of d observations in R, beginning with $R_{i}$, as shown in Figure 3a, top. We denoted the observed alignment calculated by DTW algorithms as: $Q_{j}\overset{obs}{↔}G_{k}$.

### 2.3. Design of Error Rate Metric

As described above, we proposed the error rate (ER) as a metric to quantify the quality of an alignment between two signals, i.e., how different any given alignment is from the optimal alignment. We defined ER∈[0,1], where ER = 0 indicates the optimal alignment and ER=1 indicates the worst possible alignment:(6)$$\;ER=\frac{\sum_{i=1}^{L_{R}}\left|k-j\right|\;}{\frac{1}{2}L_{Q}\left(L_{Q}-1\right)\times \left(1+d\right)-\left(L_{Q}-1\right)}$$

We quantified misalignment of each observation in R as the index difference between the observed group k and the optimal group j (see Figure 3a, middle plot, where $R_{3}$ and $R_{6}$ are misaligned). The numerator of the formula represented the sum of misalignments of observations in R. It equaled 0 when k=j, meaning that there was no difference between the optimal alignment and the observed alignment. The denominator quantified the total misalignment for all observations in R under the most severe singularity, when all observations in Q were mapped with the first observation in R and all observations in R were mapped with the last observation in Q, as shown in Figure 3b, top. We generalized this situation to down-sampled signals as shown in Figure 3b, bottom, and we calculated this possible maximum misalignment as denominator. We defined this alignment as the “worst-case scenario” since no useful relationship was found between the two signals. In this case, the numerator equaled the denominator, and thus ER=1.

### 2.4. Design of the Singularity Score

In order to measure the severity of the singularity problem, we proposed the singularity score (SS) by introducing a ratio inspired by the variance calculation for a dataset:(7)$$SS=\frac{\sum_{j=1}^{L_{Q}}{\left(m_{j}-d\right)}^{2}}{L_{Q}}$$
where $m_{j}$ is the number of observations in the observed group $G_{k}$. The down-sampling rate d represents the ideal number of links from each observation in Q, assuming the signals are evenly sampled.

In comparison with the warping path metric W used by Keogh et al. to count the length of warping path [8], the squared term in Equation (7) provided higher sensitivity to uneven dispersion of alignment pairing, i.e., when there was higher variation in the number of observations in R that were matched to a single observation in Q. For example, in Figure 3a, the middle and bottom cases had the same W but the bottom case had a higher SS because it had larger variance of $m_{j}$.

However, the singularity problem was directional when the frequencies of signals differed. This meant that the situation where M (M>1) observations from the lower frequency signal Q were matched to one observation in the high-frequency signal, R was much worse than vice versa. Therefore, when the frequencies of the signals differed, we set $m_{j}=\frac{1}{M}$ to increase the penalty for these types of misalignments. Since d was always ≥ 1, the squared term was larger when $m_{j}<1$. Figure 3b also depicts this situation, where the first two observations in Q are penalized more by setting $m_{j}=\frac{1}{3}$ in the SS.

### 2.5. EventDTW

EventDTW was designed to improve the alignment of two signals when they measure the same phenomenon with different frequencies. Considering that global signal shape- and derivative- based algorithms like sDTW and dDTW ignore key local information, we propose a DTW algorithm that takes advantage of local information to produce better overall alignments with respect to the whole signal. Inspired by the design of the Hawkes process [21], we introduce our novel alignment algorithm, EventDTW, which is based on information propagation of events in a signal. We define the events in this algorithm as any upslope or downslope. This is because the end of an upslope or the start of a downslope result in a peak, which is the prominent event that we deem to be important for alignment between signals. An important aspect to note in our current implementation of eDTW is that we have defined an “event frequency” parameter that is user-defined. In the remainder of this paper, we set this parameter to 20%. We limit the slopes to be considered as events in order to align signals according to general shape characteristics because in this way we can avoid designating noise or artifacts as significant events to align.

#### 2.5.1. Event Detection and Selection

Since the trends of a signal segment can be classified as either an upslope, a downslope, or a plateau, by our definition we denoted the types of events as up or down. There were four steps for processing event alignment, and the following is an example of selecting an upslope as an event:
Select an entire upslope with monotonically increasing observations in the signal. An upslope is defined such that it is a contiguous subset of observations starting with the lowest point and continuously increasing until there is a decrease or stagnation from one point to another.Select the key slopes. We selected the slopes with the highest 20% elevation in R based on empirical trials. Elevation is defined as the product of the number of observations in a slope and the amplitude difference between the maximum point and the minimum point of the slope.Every upslope in Q should match an upslope in R based on proximity in timestamps. If a slope in Q is not matched, it is excluded.Set the peak of the upslope as an event and propagate the information along the slope as per below.

#### 2.5.2. Event Information Propagation

We denoted the $n^{th}$ event as $E_{n}$ and the timestamps and values of each $E_{n}$ as $\left(ts_{E_{n}},\;te_{E_{n}},\;vs_{E_{n}},\;ve_{E_{n}}\right)$, where the starting point of a slope was the observation with maximum amplitude, and the end point was the observation with minimum amplitude, as shown in Figure 4a. The event information propagation depended on the difference between the timestamp of an observation on a slope and $ts_{E_{n}}$. The event information of an observation decayed as this difference increased. Instead of just using $v_{R_{i}}$ as the point-to-point distance in original DTW, we denoted that $\;V_{Ri}$ was the vector describing the information of $R_{i}$, including three dimensions: $v_{R_{i}}$ was the amplitude value, and $\;I_{up,\;R_{i}}$ and $I_{down,\;R_{i}}$ represented the event information of $R_{i}$ belonging to an upslope or a downslope, respectively. The vector representation of R was as follows:(8)$$V_{Ri}=\left(v_{R_{i}},I_{up,\;R_{i}},I_{down,\;R_{i}}\right)i\in \left\{1,2,\ldots ,L_{R}\right\}$$

The decay process of the event information was exponential. We set $I_{R_{i}}=0.1$ at the end point of an event ($t_{R_{i}}=te_{E_{n}}$) and $I_{R_{i}}=1$ at the start point ($t_{R_{i}}=ts_{E_{n}}$). The information value at the event end point, 0.1, was chosen arbitrarily between 0 and 1 as this information value could not be close to 1 because then there would not be a sufficient difference in information between the maximum and the minimum point of an event; it also could not be too close to 0 because then we could not distinguish an observation in a significant event from an observation that was not in an event.
(9)$$I_{\;R_{i}}=\{\begin{array}{c} e^{\ln\left(0.1\right)\frac{\left|t_{R_{i}}-ts_{E_{n}}\right|}{\left|te_{E_{n}}-ts_{E_{n}}\right|}\;up\left(ts\leq t_{R_{i}}\leq te\right)\;OR\;down\left(te\leq t_{R_{i}}\leq ts\right)}\\ \;\\ 0\;otherwise \end{array}$$

The event information propagation of Q was the same as that of R. Our eDTW found the optimal alignment between Q and R by calculating minimum Euclidean distance of each $V_{Ri}$ and $V_{Qi}$.

Both slope detection and key slope selection required looping over values of R and Q. Event matching and event information propagation had a constant computational cost regardless of the spatiotemporal complexity. The spatiotemporal complexity of the event information calculation process described in Section 2.5.1 and Section 2.5.2 were $O\left(L_{R}+L_{Q}\right)$. After we defined the event information, the process of finding the optimal alignment was the same as typical DTW, and its spatiotemporal complexity was $O\left(L_{R}\cdot L_{Q}\right)$. The total spatiotemporal complexity of eDTW was $O\left(L_{R}\cdot L_{Q}\right)$, which was the same as DTW.

For scaling EventDTW, several components for the methodology could be optimized. For the slope calculation, event matching, and event information propagation, parallelization could be applied for scaling since the individual computations were not dependent. However, for the key slope selection step, synchronization was needed because the key slopes were selected based on the comparison of all detected slopes, and this did not enable parallelization. The standard DTW method could be optimized to mine massive datasets, and since EventDTW is calculated using standard DTW with the inclusion of event information, EventDTW is similarly scalable [22].

## 3. Results

### 3.1. Results of Alignments on the University of California, Riverside (UCR) Time Series Classification Archive 

In order to compare the performance of eDTW against existing state-of-the-art DTW algorithms, we synthetically warped a set of 84 publicly available time series datasets (https://www.cs.ucr.edu/~eamonn/time_series_data/) by introducing a localized time shift and an amplitude distortion to the signal amplitudes. Perfect alignment of this warped time sequence to the original, optimal sequence would give an exact one-to-one index correspondence, i.e., a 0% error rate. The error rate compares the alignment resulting from using a DTW algorithm against a perfect alignment that we calculated in II.B, and outputs a proportion score of error.

We tested the performance of DTW, dDTW, sDTW, and eDTW using SS and ER on a gold-standard open-access public dataset, including 84 different signals collected from various scenarios. We preprocessed each dataset using Gaussian smoothing kernels and standardized the signal amplitude. We then generated synthetically warped and down-sampled companion signal Q and then calculated the SS and ER for each alignment with the reference signal R generated by each of the four algorithms. For 25% of all 84 cases, eDTW outperformed DTW, sDTW, and dDTW in terms of both SS and ER. Compared with DTW, dDTW, and sDTW, the spatiotemporal complexity of eDTW was the same as that of DTW and dDTW [8], and slightly better than that of sDTW [11].

Table 1 (based on Appendix A) shows the percentage of cases in which a particular DTW performs best in terms of SS and ER. When the peaks of both signals are clearly defined and correctly detected, eDTW significantly mitigates the singularity problem and provides a higher accuracy (Figure 4b). By observing the alignment results of the 84 datasets, we discovered a trend in which eDTW performed worse than DTW when the sequences were too sparse and when the sequences were not smooth, (i.e., when tiny fluctuations existed throughout the sequences) (Appendix A). On the other hand, both eDTW and DTW outperformed both sDTW and dDTW based on the error rate and singularity score for all 84 datasets.

### 3.2. Results of Alignments on ECG/PPG Heart Rate Datasets

We also applied our methods in aligning two real-world biomedical signals that we collected from simultaneous electrocardiogram (ECG) and photoplethysmogram (PPG) sensing using wrist and chest measurement modalities. ECG is a test that measures the electrical activity of the heart. PPG is an optically obtained plethysmogram that can be used to detect blood volume changes in the microvascular bed of tissue. In this study, we experimented with a dataset that we collected [22] consisting of heart rates derived from ECG signals measured at a frequency of 1000 Hz by a 3-lead ECG (Bittium Faros 180, Bittium, Inc., Oulu, Finland) and PPG-based heart rates sampled at a dynamic sampling rate of approximately 0.2 Hz, gathered with the Apple Watch 4 (Apple Inc., Cupertino, CA, USA).

We calculated heart rate (HR) from the ECG using the standard RR intervals, which is the time elapsed between two successive R-waves of the electrocardiogram, using Kubios HRV Premium (version 3.3). Heart rate values in units of beats per minute at each timestamp were calculated by taking the inverse of the RR interval and multiplying by 60. We compared methods to align the processed heart rate signals from the ECG and the Apple Watch 4. We preprocessed both the ECG and PPG heart rate data using Gaussian smoothing kernels and standardized the signal amplitude. We then aligned the PPG and ECG signals using four different algorithms, DTW, dDTW, sDTW, and eDTW, and calculated the SS for each.

In the experiments applying DTW algorithms on the biomedical dataset STEP [23], we observed that eDTW outperformed DTW, sDTW, and dDTW in 76% of the 38 pairs of signals in terms of singularity score (Table 2 calculated based on Appendix A). We were not able to calculate ER because there was no way to obtain the optimal alignment for these real-world signals.

## 4. Discussion

The evaluation of DTW algorithms has proven to be a significant challenge. With the two evaluation metrics proposed here, we have enabled for the first time a quantitative method to evaluate and compare DTW algorithm performance. Further, our results demonstrate improved signal alignment using our novel algorithm, EventDTW, as compared to the original DTW algorithm for a significant number (36.9%) of cases in the 84 gold-standard signal processing datasets. EventDTW is also strictly superior to the dDTW and sDTW algorithms in addressing the singularity problem when the signals to be aligned are of different frequencies. 

However, as can be seen in Figure 1a, the alignment of the signals in a ’plateau’ area still tend to result in a singularity problem for four DTW methods investigated in this paper. We expect this to be mitigated by including plateau information as a weight in a modified version of the eDTW algorithm. Furthermore, our event detection method has significant room for improvement, since it is currently not guaranteed to perfectly find and match events from two different signals. 

We believe that eDTW has the potential to significantly outperform other DTW algorithms in specific circumstances, one of which is when the two signals we attempt to align are measured at different frequencies and have distinguishable events. However, signals of different characteristics may require different DTW algorithms for optimal alignment. The more pronounced the events in the signals are, the likelier eDTW is to find the global optimum alignment. We believe that eDTW is flexible and can be adjusted to different situations and may produce the optimal alignment under nearly all circumstances if the choice of the events is carefully delineated prior to performing alignment with eDTW. Future work will involve designing and testing a signal characteristic metric to determine the optimal DTW algorithm, or algorithm parameters will optimize alignment for signals of different characteristics.

## 5. Conclusions

In this article, we compared existing DTW algorithms and illustrate their limitations with regard to the singularity problem and accuracy in aligning signals of different sampling frequencies. We detailed the construction of the error rate and the singularity score metrics for evaluating and comparing DTW algorithm performance. We then demonstrated the significant potential of EventDTW in addressing singularity problems and producing accurate alignments by applying eDTW to both synthetic time sequences as well as real-world biomedical signals. While several limitations still exist and future steps of research and analysis are needed, the eDTW algorithm outperforms existing DTW algorithms for aligning real-world biomedical signals and addresses a wide need in this application area.

## Figures and Tables

**Figure 1 sensors-20-02700-f001:**
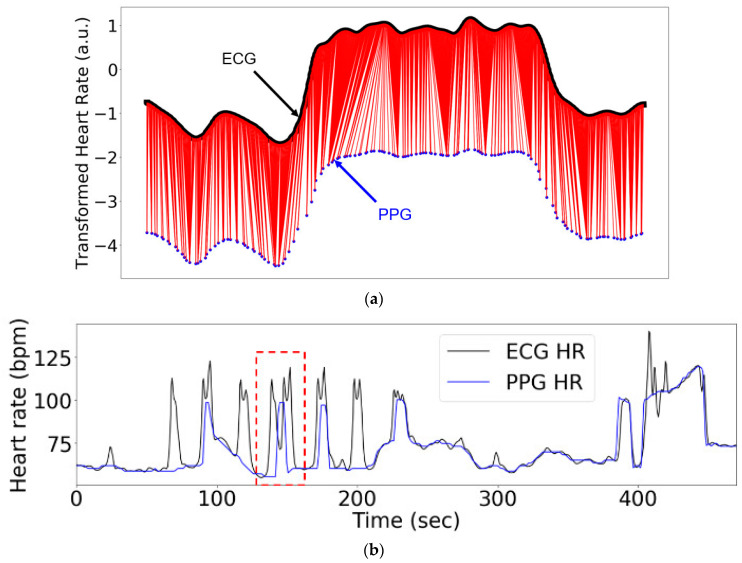
Real multi-modal biomedical signals underlie the need for both an improved alignment methodology and an improved metric to evaluate the resulting alignment: (**a**) Heart rate measured with ECG (black, ~1.26 Hz) and photoplethysmogram (PPG) (blue, ~0.2 Hz); (**b**) an example of warping and distortion observed when comparing the heart rates measured by ECG (black, ~1.26 Hz) and PPG (blue, ~0.2 Hz).

**Figure 2 sensors-20-02700-f002:**
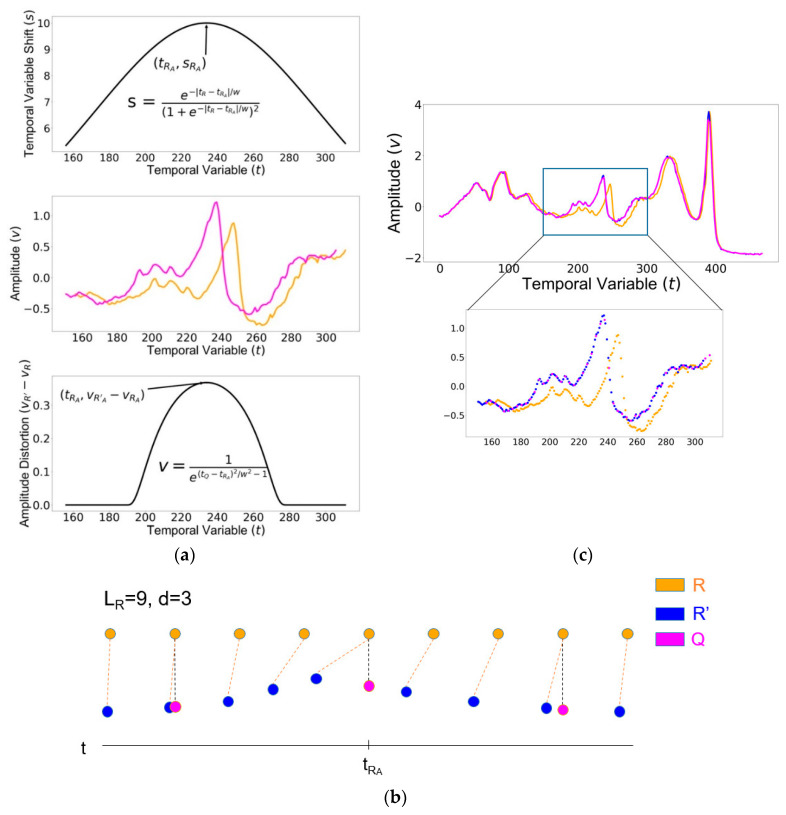
Generating the warped and down-sampled companion signal Q from the reference signal R: (**a**) The temporal variable shift and amplitude distortion of the exemplary signal; (**b**) a diagram of the companion signal generation process; (**c**) an example of the original signal R, the intermediate signal $R^{\prime }$, and the warped and down-sampled companion signal Q.

**Figure 3 sensors-20-02700-f003:**
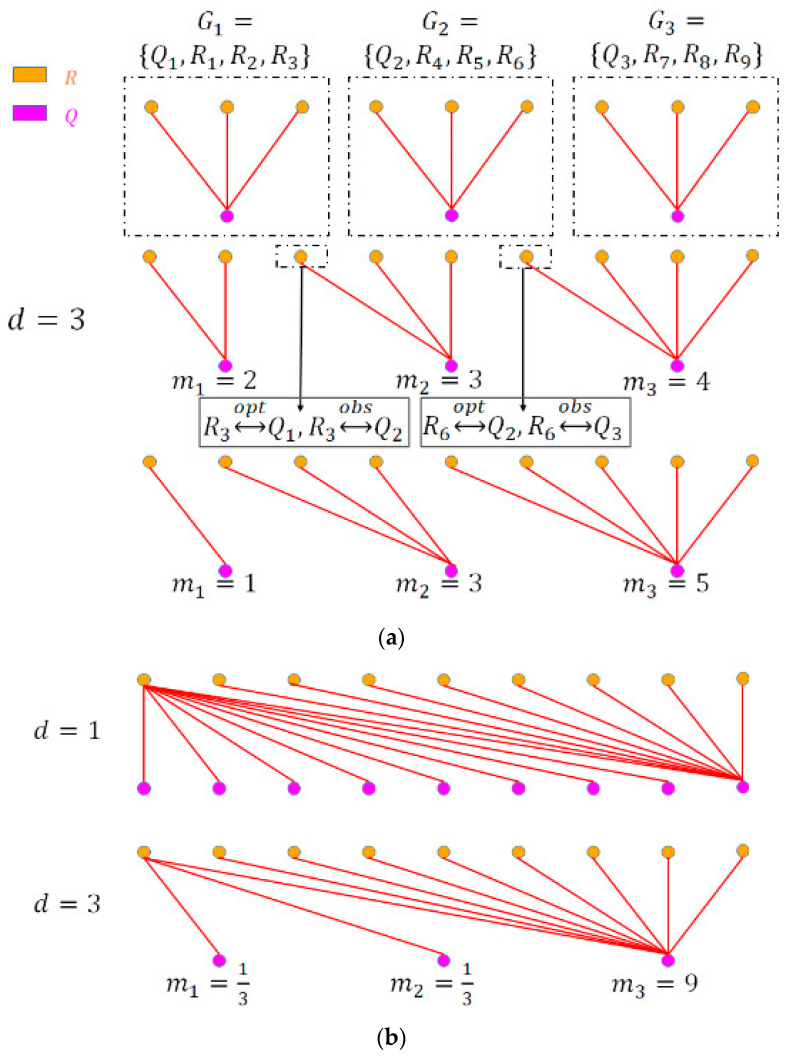
Calculation of the optimal alignment, error rate, and singularity score: (**a**) An example of alignment; (**b**) the alignment with the most severe singularity problem.

**Figure 4 sensors-20-02700-f004:**
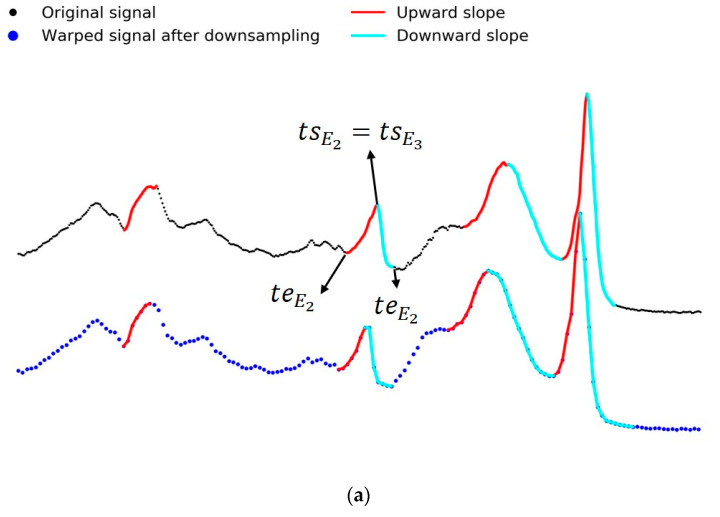
An example of application of four dynamic time warping (DTW) algorithms to the BEEF signal from UCR dataset [20] in which EventDTW (eDTW) outperforms DTW, derivative DTW (dDTW), and shape DTW (sDTW). (**a**) The event information propagation process, and (**b**) the alignment after applying four methods to the signals in (**a**).

**Table 1 sensors-20-02700-t001:** Algorithm performances in the UCR Time Series Classification Archive.

Algorithm	Percent of Best Algorithm by Singularity Score	Percent of Best Algorithm by Error Rate
eDTW	42%	37%
DTW	32%	49%
sDTW	7%	4%
dDTW	19%	11%

**Table 2 sensors-20-02700-t002:** Algorithm performances in biomedical dataset [22].

Algorithm	Percent of Best Algorithm by Singularity Score
eDTW	76%
DTW	0%
sDTW	0%
dDTW	24%

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
