# Peer review of "EventDTW: An Improved Dynamic Time Warping Algorithm for Aligning Biomedical Signals of Nonuniform Sampling Frequencies"

_sensors, 2020, doi:10.3390/s20092700_

Round 1
Reviewer 1 Report
This paper proposed a DTW algorithm called EventDTW. Although some results seems effective, I cannot find the codes from the website: https://github.com/Big-Ideas- 102 Lab/DBDP/tree/master/DigitalBiomarkers-Preprocessing. To guarantee the results are effective and helpful to the readers, I think it is very necessary to include the original codes in the paper since there are so many figures.Author Response
We apologize for this oversight and have included the original code for the paper in the GitHub repository: https://github.com/Big-Ideas-Lab/DBDP/tree/master/DigitalBiomarkers-Preprocessing/Signal-Alignment
We also provide instructions on how to regenerate each figure and table in the README.
Reviewer 2 Report
The authors propose an augmentation of dynamic time warping (DTW) to mitigate limitations of the method working with similarity search on datasets with different recording frequencies.
The manuscript is well-written and easy to read.
The idea is pretty interesting and well worth considering for publication. However:
- The authors do not talk about lower-bound and upper-bound characteristics of the proposed event DTW method.
- The authors do not talk about time and space complexity of the proposed event DTW method, in a side by side comparison against the state-of-the-art DTW methods.
- Also, the proposed event DTW method is compared against the state-of-the-art DTW methods based on two measures of singularity score and error rate. However, there is no statement about scalability of the proposed event DTW method and how good the method would work at scale?
- Can you please add more real-world datasets (not contrived as the datasets in UCR or UCI archives) on top of ECG / PPG datasets?
I look forward to seeing the revised version of the manuscript.
Reviewer 3 Report
In this manuscript, authors demonstrate an advanced dynamic time warping (DTW) algorithm to align signals of uneven sampling frequencies. The algorithm builds the mapping matrix by considering distinguishable “events” in signals. In addition, authors propose two metrics to evaluate the performance of the algorithm and further compare it to other existing state-of-the-art counterparts. The event-based DTW shows superiority over the counterparts regarding the singularity issue.
From my point of view, this is competent work. The presentation of the paper is clear. The paper is written such as to be understandable for casual readers. I believe this proposed algorithm is of broad interest for IoT where data synchronization is important. Besides, the two metrics pave the way for later researches.
I conclude that this manuscript reaches the criteria of presentation on Sensors for the reasons listed above. I recommend publishing it.
Author Response
We would like to thank Reviewer 3 for their thoughtful comments. No items to be addressed.
Round 2
Reviewer 1 Report
I agree to publish this revised version.
Author Response
We would like to thank Reviewer 1 for their suggestions. No items to be addressed.
Reviewer 2 Report
I find this paper to be interesting, novel, actionable and worthy of publication.
I would like the authors to address the following, at a minimum a few lines to say why this is not a solution.
1) According to “myth 1” of [a], we can simply reinterpolate the two time series being compared, then compare them as normal. If we do this, there is no loss of accuracy.
To make this clear, suppose we have (in matlab)
heartbeatA % sampled at 1,000hz
heartbeatB % sampled at 100hz
You could simply….
heartbeatA = heartbeatA(1:10:end); %% this is a hack that works, you can use interp1
Now you can do DTW(heartbeatA , heartbeatB )
2) (see [b] fig 2). A related trick, suppose you have…
heartbeatA % sampled at 1,000hz
heartbeatC % sampled at 1,000hz
heartbeatX % sampled at 337hz
heartbeatY % sampled at 337hz
And you have already measured DTW(heartbeatA , heartbeatB) and separately DTW(heartbeatX , heartbeatY) in the original resolution.
You now want to compare the comparisons, to see which pair is closest. However, because of the different sampling rates you cannot…
You CAN make them commensurate, all you have to do is divide each by the square root of the reciprocal of the original data’s length.
3) Most of the images are low quality bitmaps, not vector graphics. In fig 4, if you skipped every second line in the alignment plots, these would be easier to interpreted.
[a] Ratanamaha. Three Myths about Dynamic Time Warping. In proceedings of SIAM International Conference on Data Mining (SDM '05), pp. 506-510.
[b] Matrix Profile X: VALMOD - Scalable Discovery of Variable-Length Motifs in Data Series. Michele Linardi SIGMOD 2018.
